# SMP Production in an Anaerobic Submerged Membrane Bioreactor (AnMBR) at Different Organic Loading Rates

**DOI:** 10.3390/membranes10110317

**Published:** 2020-10-30

**Authors:** Sandra C. Medina, Nataly Zamora-Vacca, Hector J. Luna, Nicolas Ratkovich, Manuel Rodríguez Susa

**Affiliations:** 1Environmental Engineering Research Center (CIIA), Department of Civil and Environmental Engineering, Universidad de Los Andes, Cra. 1 #18a 12, Bogotá 111711, Colombia; sandra.medina@kaust.edu.sa (S.C.M.); hectorlunaw@uan.edu.co (H.J.L.); manuel-r@uniandes.edu.co (M.R.S.); 2Water Desalination and Reuse Center (WDRC), Biological and Environmental Science & Engineering (BESE), King Abdullah University of Science and Technology (KAUST), Thuwal 23955-6900, Saudi Arabia; 3Escuela de Ciencias, Agrícolas, Pecuarias y del Medio Ambiente, Universidad Nacional Abierta y a Distancia, Calle 14 Sur # 23 - 14, Bogotá 551015, Colombia; 4Facultad de Ingeniería Ambiental, Universidad Antonio Nariño, Calle 22 Sur # 12D - 81, Bogotá 111511, Colombia; 5Department of Chemical and Food Engineering, School of Engineering, Universidad de Los Andes, Bogotá 111711, Colombia

**Keywords:** anaerobic membrane bioreactor (AnMBR), membrane fouling, soluble microbial products (SMPs), wastewater treatment

## Abstract

Anaerobic membrane bioreactors (AnMBRs) have demonstrated an excellent capability to treat domestic wastewater. However, biofouling reduces membrane permeability, increasing operational costs and overall energy demand. Soluble microbial products (SMPs) that build up on the membrane surface play a significant role in the biofouling. In this study, the production of SMPs in a 32 L submerged AnMBR operated at three different organic loads (3.0, 4.1 and 1.2 kg chemical oxygen demand (COD)/m^3^d for phases 1, 2 and 3, respectively) during long-term operation of the reactor (144, 83 and 94 days) were evaluated. The samples were taken from both the permeate and the sludge at three different heights (0.14, 0.44 and 0.75 m). Higher production of SMPs was obtained in phase 2, which was proportional to the membrane fouling. There were no statistically significant differences (*p* > 0.05) in the SMPs extracted from sludge at different heights among the three phases. In the permeate of phases 1, 2 and 3, the membrane allowed the removal of 56%, 70% and 64% of the SMP concentration in the sludge. SMPs were characterized by molecular weight (MW). A bimodal behavior was obtained, where fractions prevailed with an MW < 1 kDa, associated with SMPs as utilization-associated products (UAPs) caused fouling by the pore-blocking mechanism. The chemical analysis found that, in the SMPs, the unknown COD predominated over the known COD, such as carbohydrates and proteins. These results suggest that further studies in SMP characterization should focus on the unknown COD fraction to understand the membrane fouling in AnMBR systems better.

## 1. Introduction

Anaerobic membrane bioreactors (AnMBRs) are a promising technology for wastewater treatment [1,2]. The main advantage of this technology over conventional membrane bioreactor (MBR) systems is the decoupling between hydraulic retention time (HRT) and sludge retention time [3], washout prevention [2], and improvement in effluent quality [2,4]. Additionally, there is evidence that the energy balance for fouling control can be positive for AnMBRs [1,5]. Shin and Bae [1] found that the energy demand for a pilot-scale AnMBR was in the range from 0.04 to 1.35 kWh/m^3^, which is less than the requirement for an MBR. The application of AnMBRs includes low-strength wastewater (275–1462 mg chemical oxygen demand (COD)/L; 0.4–2.5 kg COD/m^3^d) [1,6] and high-strength wastewater in the confectionery (25,000–60,000 mg COD/L) [7,8], sugar (16,706 mg COD/L) [9], dairy (48,200 mg COD/L) [10], wine (3.4 kg COD/m^3^d; 6752 mg COD/L) [11], and beer industries (2–10 kg COD/m_3_d). These examples show the capability of AnMBRs operating under a wide range of organic loading rates (OLRs).

The main challenge for a more comprehensive implementation of AnMBRs is the fouling phenomenon, which has been studied in recent years. According to Spagni et al. [12], higher fouling in AnMBRs is obtained compared to MBRs. The primary membrane fouling mechanisms were attributed to cake formation and pore-blocking for aerobic and anaerobic MBRs, respectively [13]. The adhesion/deposition of extracellular polymeric substances (EPSs) and soluble microbial products (SMPs) on the membrane have been associated as critical precursors of membrane fouling [14]. SMP production, in particular, appears to be one of the essential factors to understand the fouling mechanism in AnMBRs [9]. The identification of SMPs in the supernatant allowed knowing the metabolic pathways of their production. SMPs are difficult to degrade, presenting effects on the health and the environment [15]. These SMPs require identification to improve after-treatment processes.

SMPs can be classified into utilization-associated products (UAP) and biomass-associated products (BAP) [16]. The UAP has a molecular weight (MW) of less than 1 kDa and is biodegradable. The BAP has an MW greater than 10 kDa, with hydrophilic characteristics, and the BAPs are less biodegradable [13]. SMPs are also a crucial part of effluent COD, which make up residual COD by a higher percentage [17,18,19]. The characterization of residual COD is complex and has generally been performed by collective methods such as molecular weight distribution [14,15,20] and biodegradability to classify SMPs [21]. Aquino and Stuckey [17] characterized anaerobic reactor effluents (CSTRs and MBRs) by molecular weight distribution, finding a high percentage of SMPs in the range of >10 kDa, showing the importance of relatively low molecular weight SMPs. In terms of SMP chemical characterization, it is still not clear under which conditions carbohydrates or proteins may be more critical for membrane fouling [22,23]. Other studies have shown that unknown COD might have a crucial role in some wastewater treatment effluents besides proteins and carbohydrates [24]. Further analysis of the unknown COD by MW distribution is needed to understand its role in membrane fouling better.

Production of SMPs has been correlated to membrane fouling under different OLRs during AnMBR operation. Chen et al. [14] studied the influence of SMPs on low-strength wastewater fouling (i.e., OLRs of 0.7 and 1.4 kg COD/m^3^d). In this study, it was found that for the 0.7 kg COD/m^3^d, the fouling was attributed to the pore-blocking mechanism induced by SMP production, while for the OLR of 1.4 kg COD/m^3^d, the fouling was attributed to the cake formation mechanism induced by EPS production. On the other hand, Santos et al. [9] conducted a study on the influence of SMPs on industrial wastewater fouling for three OLRs (2.5, 4.7, and 6.0 kg COD/m^3^d), decreasing the HRT. It was found that for OLRs of 2.5 and 6.0 kg COD/m^3^d, the fouling mechanism was pore-blocking, and for the 4.7 kg COD/m^3^d, it was due to the cake formation mechanism. Balcıoğlu et al. [7] analyzed the variation in OLR (1.1–7.9 kg COD/m^3^d) for the wastewater of the confectionery industry. In that study, SMPs were identified as the leading cause of pore-blocking when operating at a low OLR, while EPSs were correlated to cake formation during high OLR operation. Given the impact of SMP in the membrane fouling of AnMBR reactors for industrial wastewater, this study aims to evaluate the SMP production in a submerged AnMBR under relatively high OLRs of 1.2, 3.0, and 4.1 kg COD/m^3^d conditions. The SMP analysis was performed using MW distribution with the characterization of carbohydrates and proteins. The sampling point at a different height in the reactor is included as a variable, considering previous studies [25]. This research is one of the first studies focusing on the characterization of MW distribution in an AnMBR at different heights and OLR by identifying the role of SMPs in membrane fouling.

## 2. Materials and Methods

### 2.1. Reactor Description

A lab-scale AnMBR with submerged ultrafiltration membrane configuration of ascending flow was used in this study (Figure 1). The AnMBR is made of stainless steel with a volume of 31.4 L, 1 m height, and 0.20 m of diameter. The reactor has three sampling points for the sludge (P1, P2, and P3) located at different heights (0.145 m, 0.445 m, and 0.75 m, respectively). The permeate outlet point (P4) is located on the top.

The seed sludge of the reactor came from a wastewater treatment plant of a fruit juice company. The sludge was continuously recirculated from the middle to the bottom of the reactor. A heating jacket controlled the temperature under a mesophilic condition (35 ± 3 °C).

### 2.2. Ultrafiltration Membrane

A U-shaped hollow fiber membrane module was constructed in-house using an UltraPES polyethersulfone (PES) hollow fiber membrane (3MTM, Los Alamitos, CA, USA). The UltraPES capillaries had 0.7 mm of inside diameter and 0.01 µm pore size. The effective filtration area of the membrane was 0.38 m^2^. The membrane was in the upper part of the reactor. The operation mode of the membrane was in cycles of 10 min of suction, 1 min of relaxation, and 4 min of backwashing.

### 2.3. Reactor Feed at Different OLR

The reactor feed consisted of synthetic wastewater that contained glucose, peptone, and yeast extract as a carbon source. Details of the feeding solution can be found in Luna et al. [25]. Three consecutive runs of the reactor were performed. In the first one (phase 1), the reactor was fed with the synthetic wastewater at 6 g COD/L concentration, equivalent to an OLR of 3.0 ± 0.8 kg COD/m^3^d. This phase 1 lasted 144 days. In the second phase, the feed concentration was increased to 9 g COD/L, equivalent to an OLR of 4.1 ± 0.8 kg COD/m^3^d. This phase 2 had a duration of 83 days. In phase 3, the concentration of the feed was decreased to 3 g COD/L, and the operation lasted 94 days with an OLR of 1.2 ± 0.3 kg COD/m^3^d.

During each phase, backwash cleaning was performed when the permeate suction pressure reached a maximum value of 32 kPa. A chemical wash with NaOH and HNO3 at 5% *w/v* was performed to the membrane by the end of phase 1. For phase 3, a new ultrafiltration membrane module was used due to the deterioration of the membrane during the first two phases.

### 2.4. Sample Treatment and Frequency

Two types of sampling, depending upon the frequency, were performed: regular and special. Periodic samples were taken every seven days for phase 1, and every five days for phases 2 and 3. 100 mL of sludge were collected from each of the points P1, P2, and P3, and 250 mL of permeate were collected from P4. Except for the samples from P4, all samples were centrifuged at 13,000 rpm for 15 min at 4 °C. The supernatant obtained from each point and the permeate from P4 were then filtered by a 0.2 μm membrane. Chemical oxygen demand (COD), volatile fatty acids (VFAs), carbohydrates, proteins in terms of COD, total suspended solids (TSSs), and volatile suspended solids (VSSs) were measured on the filtered samples.

On the other hand, special sampling for the MW characterization of SMPs was done at the beginning, the middle, and the final part of each phase. This is every 28, 15 and 20 days for phases 1, 2, and 3, respectively. These samples had the same treatment as those for regular sampling. A higher volume of 200 mL sludge was extracted from P1, P2 and P3. A summary of the procedure and frequency of the samples is shown in Figure 2a.

### 2.5. Parameter Quantification Methods

The soluble COD was determined by reactor digestion method 8000 [26]. The VFAs were measured by the headspace solid-phase microextraction (HS/SPME) gas chromatography method, using an HP 6890 series Plus^®^ gas chromatograph with an FID detector, and a 60 m × 0.2 mm HP-INNOWas column (Agilent Technologies, Wilmington, DE, USA). 3 g of NaCl were added to 10 mL of sample. Phosphoric acid was used for adjusting the pH to 2. After equilibration at 60 °C for 10 min, a 75 μm Carboxen^®^-PDMS fiber (Merck, Darmstadt, Germany) was exposed to the headspace above the sample for 30 min. The fiber was finally injected in the chromatograph for analysis.

The protein quantification was performed according to Bradford’s colorimetric method [27]. Absorbance was measured with the NanoDropTM 2000/2000c spectrophotometer (Thermo Fisher Scientific, Wilmington, DE, USA) at a wavelength of 595 nm.

The carbohydrates quantification was made with the phenol/sulfuric method, based on the methodology described by Dubois et al. [28]. Absorbance was measured with a GenesysTM 10 UV spectrophotometer (Thermo Fisher Scientific) at a wavelength of 525 nm. In this same spectrophotometer, the glucose was quantified using a biosystems glucose kit [29].

SMP is calculated in two ways in terms of carbon–oxygen demand—COD (mg/L) [24].
SMP = *Sol* − *Subs* − VFA(1)
SMP = SMP*p* + SMP*c* + *Unknown* COD(2)
VFA = 1.07 × [*acetate*] + 1.51 × [*propionate*] + 1.82 × [*butyrate* + *isobutyrate*] + 2.04 × [*valeric* + *isovaleric*](3)
where: *Sol* = COD soluble (effluent filtered at 0.2 µm), *Subs* = COD substrate (glucose), VFA = volatile fatty acids in terms of COD, and SMP*p* and SMP*c* = protein and carbohydrates in terms of COD, respectively.

### 2.6. SMP Molecular Weight Distribution

The 0.2 μm filtered samples were passed through three ultrafiltration discs Ultracel^®^ (Merck, Germany) of different nominal molecular weight limit (NMWL) in a parallel configuration, as shown in Figure 2b. The NMWLs selected were 1, 10, and 100 kDa, according to the literature review. A pressure of 3 bar with industrial nitrogen was applied during the ultrafiltration in the Amicon^®^ stirred cells model 8200 (Merck, Germany).

20 mL out of 30 mL of sample were filtered in the Amicon cell. The 10 mL retained were taken to 140 mL adding Milli-Q deionized water and filtered again until a 20 mL of retentate was obtained. This retentate was sonicated with the membrane to improve the recovery of the material from the membrane surface.

The SMP in the retentates were measured for obtaining the SMP in four ranges of molecular weight: MW < 1 kDa, 1 kDa < MW < 10 kDa, 10 kDa < MW < 100 kDa, and MW > 100 kDa. The calculations based on COD mass balance are shown in Equations (4) to (7).
MW < 1 kDa = (0.03 × SMP*_sol_*) − (0.02 × SMP*_ret_* 1 kDa)(4)
1 kDa < MW < 10 kDa = 0.02 × (SMP*_ret_* 1 kDa − SMP*_ret_* 10 kDa)(5)
10 kDa < MW < 100 kDa = 0.02 × (SMP*_ret_*10 kDa − SMP*_ret_*100 kDa)(6)
MW > 100 kDa = 0.02 × SMP*_ret_* 100 kDa(7)
where SMP*_sol_* (mg COD/L) is the SMP content in the initial 30 mL sample, and SMP*_ret_* (mg COD/L) is from the 20 mL retentate obtained in each ultrafiltration; to calculate SMP, measurements of COD, VFAs, carbohydrates, and proteins were required, according to the definition given previously.

### 2.7. Theoretical Model of Membrane Fouling

The particle-free permeate through a clean membrane can be described with Darcy’s law. However, the flux deviates from Darcy’s law as fouling increases over time. Therefore, a modified Darcy’s law, as described by Rosenberger and Kraume [30] and Lee et al. [31], was implemented to determine the permeate flux. The viscosity value on the permeate was corrected as a function of the temperature.

## 3. Results

### 3.1. Reactor Behavior

Table 1 shows the general behavior of the AnMBR during its operation in the three phases. pH values were stable in the typical range in anaerobic systems [32].

Most of the influent organic matter (COD) (i.e., 93–96%) was removed in the biological process. These high COD removal percentages are consistent with Wu and Zhou [19], who reported a 96% removal in a 16 L upflow anaerobic sludge blanket (UASB) reactor fed with glucose-based synthetic wastewater. Ho and Sung [33] also reported COD removal percentages of more than 90% due to the biological process in a 4 L AnMBR. On the other hand, the membrane efficiency is high (i.e., values above 97.1%), achieving more effective removal of COD in the permeate (P4) in all three phases. In Phase 2, the increase in OLR also increased SMP, which led to membrane fouling. This association between SMP and membrane filtration resistance has been reported by several authors [34,35,36,37,38].

While the membrane accomplished a fundamental role in TSS removal, where VSSs correspond to the main fraction of TSSs (i.e., 85–89%), no statistically significant differences (*p* > 0.05) were found in the solid content (TSSs and VSSs) of the sludge at different heights of the reactor (sludge sampling P1, P2, and P3). This suggested an excellent mixing and homogeneity of the sludge during the three phases.

### 3.2. SMP Production and Composition

Figure 3a summarized the SMP concentrations found at the four sampling points from all operational phases. SMP production during phase 2 (at each sampling point) was statistically different (*p* < 0.05) compared to the concentrations obtained in phases 1 and 3. As expected, the SMP concentration in the sludge during phase 2 was the highest. At a higher OLR, microbiological metabolism is stimulated due to the availability of energy sources, and, therefore, SMP production is increased [13,18,19,21,25]. The amount of SMP was calculated using Equation (1), where the term CODSubs was equal to zero because no glucose as residual substrate was found in all samples. These results agree with previous studies from Jarusutthirak and Amy and Mesquita et al. [24,39].

On the other hand, no statistical difference was found between the SMP concentrations obtained in phases 1 and 3 (Figure 3a). However, significant differences were found between the VFAs in these phases. The VFAs in phase 1 were higher (18.2% ± 9.5%) than those in phase 3 (1.7% ± 1.3%). This could be explained by the fact that phase 3 was performed after a run with greater OLR (i.e., phase 2). In the change from phase 2 to phase 3, the microorganisms had a drastic reduction of 70% of the feed availability, creating a stressful condition where VFA was used as a substrate. The biomass decay confirmed this in phase 3 (see VSS in Table 1), stimulating VSS degradation and, therefore, higher SMP production by cell lysis [24].

Furthermore, the increasing SMP concentration in phase 2 caused more significant membrane fouling. Weekly hydraulic cleaning of the membrane was needed during phase 2 to keep the permeate suction pressure under 32 kPa. This is three times the hydraulic cleaning frequency required for phases 1 and 3, demanding more maintenance time and operational cost. Chemical cleaning could not recover the membrane after phase 2. For phase 3, a new membrane was placed. This positive association between SMP and filtration resistance in the membrane has been reported by several authors [35,38,40]. In Figure 3a, it is possible to evidence a reduction of SMP concentration from P3 to P4 in all operational phases. This reduction was 56%, 70%, and 64% for phases 1, 2, and 3, respectively. The increasing membrane fouling in phase 2 could be associated with the formation of the cake layer during the filtration phase, creating a dynamic membrane that reduced the porosity and, consequently, a higher rejection degree [41]. Moreover, the dynamic membrane formed could cause SMP consumption to supply the substrate deficit in this zone [42].

For each of the phases, no statistical differences (*p* > 0.05) of SMP concentrations in the sludge were found at different reactor heights (P1, P2, and P3). Furthermore, Figure 3a shows a slight tendency of higher SMP concentration in P3 from all operational phases. This indicates an SMP accumulation in the upper zone due to membrane retention [38]. These results differ from those in the study by Luna [25]. They found higher SMP at the bottom of the reactor, associated with higher biomass concentration in this area. In Luna’s study [25], a clear stratification of solids throughout the reactor height was observed. This was not the case for this study. The SMP concentration in the sludge coincides with the homogeneity of VSS/TSS percentages, which remained relatively invariant at different heights.

Proteins, carbohydrates, and the unknown COD fraction are part of the SMP definition given by Equation (2). Chemical characterization results are shown in Figure 3b. The unknown COD fraction represented most of the SMP, with percentages between 81% and 90% for all sample points in all the operational phases. These results are consistent with the observations of Mesquita et al. [24], who concluded that most of the SMP does not seem to be produced by soluble EPS but, due to the complexity of anaerobic biochemistry, it remains an unknown fraction. Kunacheva et al. [15] standardized the method to identify the unknown COD for low MW in AnMBR and found 256 compounds; 134 of these compounds were not identified, and the rest were classified as alkanes, alkenes, alcohols, phenols, and esters, among others. Unknown COD compounds are highly carcinogenic, toxic, and are precursors to the byproducts of chlorination. The characterization of this fraction is essential for the reuse and reclamation of treated water [43,44].

Protein concentrations of all effluent samples in phases 1 and 3 were close to the detection limit of 5 mg/L. Such low protein concentrations have been reported previously by Le-Clech et al. [45]. In phase 2, protein concentrations were comparable to carbohydrate values. This increase of protein in phase 2 agrees with the Kimura et al. results [46], where an increasing food/microorganism ratio changes the nature of the membrane foulant to be more proteinaceous. Another explanation for this result is high sludge retention time (SRT), which causes cellular decay [13].

The SMP difference between P3 and P4 for all three phases shows that the membrane can retain or degrade some SMPs, significantly decreasing the percentage of unknown COD. One possible cause that may explain this process is the membrane fouling caused by the pore blockage of these organic compounds that make up the unknown COD, both BAPs and UAPs, as proposed in the fouling model described by Ni et al. [13].

### 3.3. SMP Molecular Weight Distribution

The molecular weight distribution results at three different moments of each operational phase are presented in Figure 4. The synthetic wastewater used as feed showed an expected MW distribution where most of the CODs (i.e., 82.7%) have an MW smaller to 1 kDa, followed by 12.2% in the 1 kDa–10 kDa range, 3.1% between 10 kDa and 100 kDa, and 1.9% higher than 100 kDa.

The ultrafiltration performed at the beginning of each phase (Figure 4a,d,g) showed the lowest MW range (<1 kDa) as the main fraction of SMP with percentages between 46% and 66%. Several researchers have observed such predominance of the smallest MW fraction when the operation time is up to 40 days [21,47,48,49]. It is possible to identify the smallest SMP fraction as UAP because microbiological products are released for environmental adaptation at the beginning of the operation.

As the operation continues, the smallest MW fraction is less predominant, giving space to the intermediate and the higher MW fraction (Figure 4c,f,i). The intermediate fraction of MW between 10–100 kDa was the least representative in phase 2. These results suggest the presence of a bimodal MW distribution of SMPs (i.e., MW < 10 kDa and MW > 100 kDa) during the operation at the highest OLR, similar to the results of Aquino and Stuckey [17], Magbanua and Bowers [50], Jarusutthirak and Amy [39], and Meng et al. [36]. According to Aquino et al. [20], this behavior in AnMBR can be explained by the degradation of high molecular weight SMP by acclimatized biomass.

In phase 1, the intermediate fraction became important at every sampling point, and by the end of the run (day 144 in Figure 4c), the smallest MW fraction decreased to 24% (see P2) of the total SMP. SMP with higher MW (i.e., >10 kDa) is accumulated at an increasing operational time due to its non-biodegradable properties [13,17], and consequently, they become the most important fraction of soluble organic matter [13].

In phase 3, the prevalence of MW < 1 kDa fraction at the beginning of the operation was not completely clear compared to the other phases. Furthermore, the MW > 100 kDa fraction obtained significant percentages, such as 38% and 32% in P1 and P2 (Figure 4g). The presence of the large MW fraction in phase 3 is a consequence of the dramatic drop of a third of the OLR in phase 2, and it was the one that had the longest solids retention time, which leads to cell lysis and the generation of higher MW SMP. The reduction in food availability causes cell lysis and SMP accumulation with MW > 100 kDa [50]. In this way and likewise phases 1, at the end of phase 3, MW > 1 kDa fraction represented the main part of SMP.

Figure 4 also shows a reduction in the fraction of high MW (>100 kDa) when it passes from point P3 and P4 for all phases, which can be attributed to the membrane being able to retain this fraction physically due to either mechanism—cake formation or pore-blocking.

### 3.4. SMP Composition of MW Fractions

The composition of each MW fraction in terms of unknown COD, carbohydrates, and proteins was identified at the beginning and the end of each operation phase. The results of SMP composition in each MW fraction are presented in Figure 5. Proteins are not included in Figure 5 because they were under the detection limit (i.e., 5 mg/L). Figure 5a shows that out of the 61% correspond to the fraction of the SMP with MW < 1 kDa in P1 (based on Figure 4a), 57% were identified as unknown COD, and the remaining 4% were carbohydrates. From Figure 5, it was possible to determine that the unknown COD (which was defined as the main fraction of total soluble COD) has a wide MW distribution in the three operation phases. At the same time, carbohydrates are present in all MW fractions, contributing to SMP to a lesser extent. Similar to these results, Mesquita et al. [24] found low carbohydrates relevance from a MALDI–TOF MS (matrix-assisted laser desorption/ionization–time-of-flight mass spectrometry) performed for MW between 20 kDa and 80 kDa.

According to SMP molecular weight distribution analysis, the lowest MW fraction (<1 kDa) predominated at the beginning of the operation in all phases, but the higher molecular weight fractions became important over time. These results suggested that SMP with higher MW (BAP) was accumulated due to its relative refractory characteristics and minor degradability. A significant presence (up to 38%) of MW > 100 kDa fraction at the beginning of phase 3 was found. This was explained as a consequence of the abrupt decrease of OLR from phase 2 to phase 3.

At the beginning and the end of each phase, high MW carbohydrates fractions tend to be in less proportion in P4 than P1–P3. This indicates that the higher MW fractions are being retained in the membrane so that the fouling would be associated with BAP. Figure 5 shows that phases 2 and 3 increase the low MW fraction of P4 from P1–P3 as reactor operating time increases, resulting in higher retention of high MW fractions over time, due to fouling caused by these high MW fractions.

The fraction of unknown COD plays an essential role in both reactor and permeation, according to Kunacheva et al. [15]. In that study, the unknown COD fraction with MW < 1 kDa was quantified, allowing identification of the metabolic route to produce these compounds. These compounds were found as precursors of chlorination, with toxic characteristics for the reuse of water [43]. According to the unified theory, SMPs can be classified based on the type of bacterial metabolism in BAPs and UAPs, where BAPs have characteristics of MW > 10 kDa and UAP < 1 kDa [13], therefore, in the supernatant and in the effluent of the AnMBR-predominated SSMP in the form of UAP.

Concerning the predominance of SMP as UAP in the AnMBR, it is also possible to explain the lower production of SMP in phase 3 (Figure 3) due to the low concentration of VSS (Table 1) and lower substrate concentration compared to the other two phases. This reaffirms the importance of chemical identification of COD on the effluent wastewater treatment products. Some studies have developed techniques for identifying these compounds, such as GC/MS (gas chromatography/mass spectrometry) for SMP identification of low molecular weights and MALDI–TOF MS (matrix-assisted laser desorption/ionization–time-of-flight mass spectrometry) for high molecular weight SMPs [17,24]. The results obtained for the characterization of anaerobic effluents showed that the GC/MS technique obtained good results for identifying low molecular weight SMP. The main compounds detected in the effluent were alkanes, alkenes, and some aromatics. Wu and Zhou [19] also found acids, esters, and some aromatics in UASB effluents for different types of industrial wastewater. Finally, Mesquita et al. [24] confirmed that the largest component of SMP in anaerobic and aerobic effluents was unknown COD using the MALDI–TOF MS technique and chemical methods for the detection of proteins and carbohydrates. An extensive review of the different analytical methods for the measurement of SMP in wastewater is summarized by Kunacheva and Stuckey [44]. Kunacheva et al. [15] validated a method for the characterization and chemical identification of low molecular weight SSMP (<580 Da) for AnMBR.

### 3.5. Fouling

Figure 6 shows the calculated membrane resistance during phase 1 operation for the three sampling points of the reactor (P1, P2, P3). As noted, there was a resistance variability calculated within the reactor on the same day. Resistances were the same in P1, P2, and P3 during days 31, 128, 140, 152, and 172 and were different at one or all points on other days. This shows that the concentrations of soluble substances and VSS differ from membrane resistance behavior. For example, resistance data showed a trend during days 65, 78, and 118, where P2 recorded higher values compared to the other points; however, during different sampling days, the concentrations of COD, carbohydrates, proteins, unknown COD and SMP were similar within the reactor (i.e., no stratification existed). Therefore, non-characterized present soluble substances (COD) may have a more significant impact on filterability compared to those evaluated. This means that unknown COD accounted for about 86% of SMP, and it predominates in the low MW fraction (<1 kDa).

The resistance variation in phase 1 shows a variation at the different points within the reactor and over time. These resistances show the influence of SMP with fouling, which, for the conditions of this AnMBR, is mainly due to low MW SMP, which, according to unified theory and the bimodal molecular weight distribution model, the fouling that occurred during phase 1 can be associated with SMP as UAP. According to the fouling model described by Ni et al. [13], the mechanisms that govern fouling are pore-blocking due to SMP (BAP and UAP) and cake formation due to EPS. This model stresses that pore-blocking is mainly due to SMP as BAP due to its larger MW size compared to SMP as UAP. The resistances in Figure 6 show that there is fouling caused mainly to SMP as UAP by the results obtained in the MW distribution in Figure 5, which causes fouling by the pore-blocking mechanism. According to the above, the variation of resistances for phase 1 can be explained according to the model of unified theory [16]; to ensure a mass balance in the electron equivalent, SMPs as BAPs produced during substrate consumption can be recirculated as electron donor substrates due to their low MW and high biodegradability.

Figure 7 shows the concentration of SMP vs. membrane resistance. This highlights the complexity of the membrane fouling analysis, considering that this study focused on soluble substances, mainly SMPs as UAPs. It should be noted that the percentage of unknown COD is a significant percentage of SMPs, so there was no need to perform tests on other types of substances. The fact that some days of operation had significant differences between the resistances recorded within the reactor and that the concentrations of COD, carbohydrates, proteins, unknown COD, SMPs, and VSSs (biomass) were the same within the reactor reflects that the fouling mechanism is a highly complex process. Similar results were found by Drews et al. [40], who determined that there was no correlation between fouling and polysaccharides. Jeong et al. [51] also found that fouling is affected by SMP; however, no characterization was performed that would allow fouling to be associated with carbohydrates, proteins, or other SMP. Figure 7 shows that the resistances of SMPs are relevant at concentrations between 100–200 mg COD/L for points P1, P2, and P3, while that for SMP concentrations in the range of 500–600 mg COD/L are relevant for points P2 and P3; this may be due to the recirculation of SMPs as UAPs as electron donor substrates.

## 4. Conclusions

The SMP production did not show significant differences within the AnMBR and presented an increase in SMP concentration concerning the increase in OLR, which was correlated with the behavior of VSS. The chemical characterization showed a domain of SMP as unknown COD to the SMP as carbohydrates and proteins; in addition to carrying out the MW distribution, this condition was maintained in all the ranges evaluated. The MW distribution presented the characteristic bimodal model with a predominance of SMP in the MW < 1 kDa range. The difference in concentration of SMPs within the reactor and the effluent indicated the SMP-related fouling. For phase 1, a variation in resistance was evident over time and within the reactor, suggesting a dynamic behavior by the SMPs, regardless of EPS. The results obtained showed that the membrane fouling was related to UAP by the pore-blocking mechanism. According to the unified theory, the membrane fouling was found dynamic because UAP can be recirculated as electron donor substrates. As future work, it is necessary to perform the identification of the fraction of SMPs as unknown COD.

## Figures and Tables

**Figure 1 membranes-10-00317-f001:**
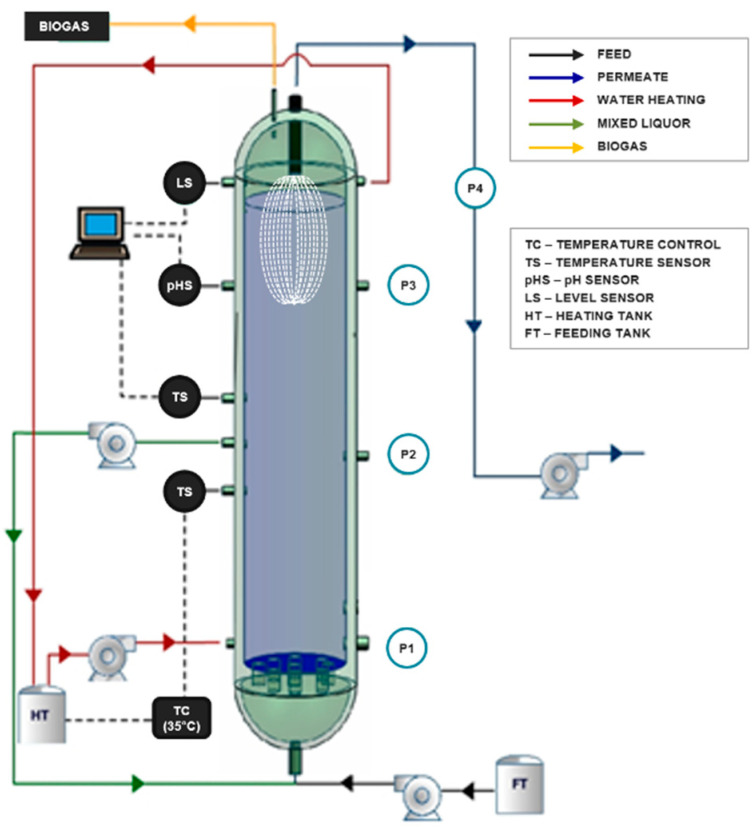
Illustration of the lab-scale submerged anaerobic membrane bioreactor (AnMBR) used in this study (adapted from Luna [25]).

**Figure 2 membranes-10-00317-f002:**
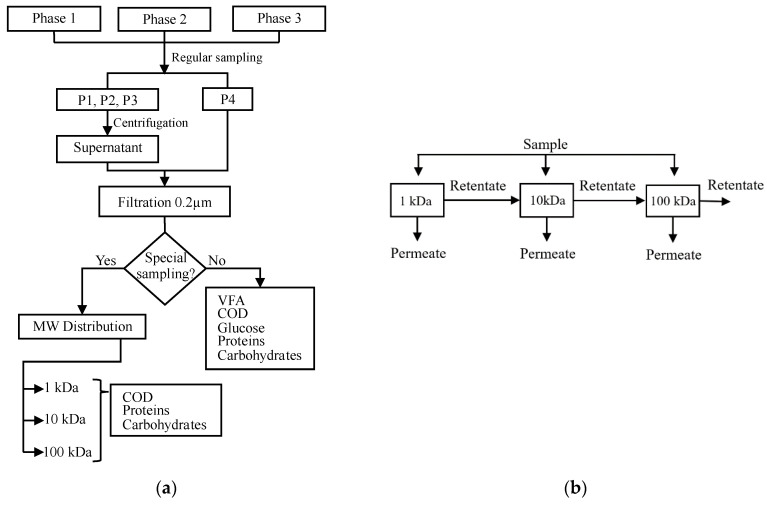
(**a**) Sampling methodology followed during the three phases. (**b**) The parallel configuration of Amicon^®^ stirred ultrafiltration cells used for soluble microbial product (SMP) molecular weight (MW) distribution.

**Figure 3 membranes-10-00317-f003:**
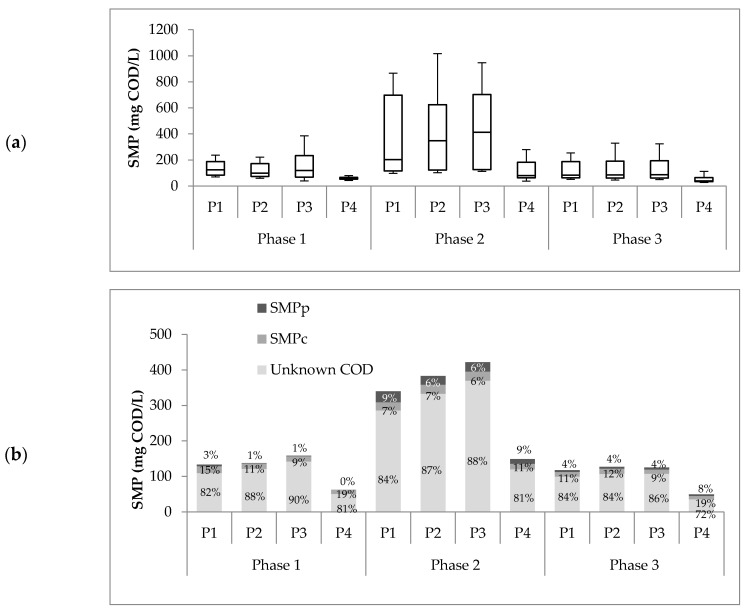
(**a**) SMP concentrations at the four sampling points from all operational phases. (**b**) Average distribution of SMPs into proteins, carbohydrates, and unknown fraction.

**Figure 4 membranes-10-00317-f004:**
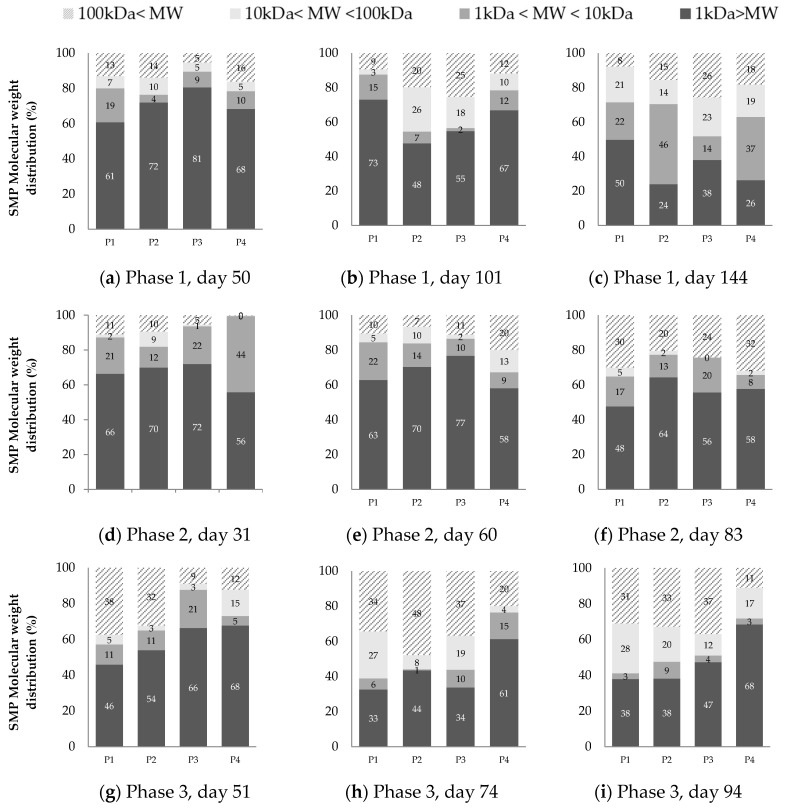
SMP molecular weight distribution. (**a**–**c**) correspond to phase 1; (**d**–**f**) to phase 2; and (**g**–**i**) to phase 3.

**Figure 5 membranes-10-00317-f005:**
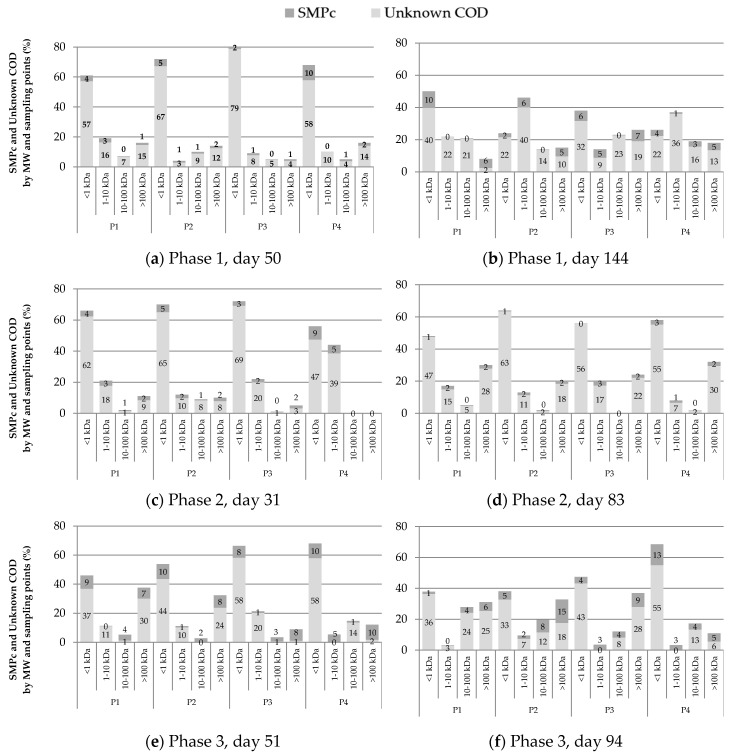
SMP percentages as unknown chemical oxygen demand (COD) and carbohydrates according to molecular weight distribution at each sample point. (**a**,**c**,**e**) correspond to the first ultrafiltration performed (day 50, 31, and 51 for phases 1, 2, and 3, respectively), and (**b**,**d**,**f**) are the ultrafiltration done at the end of operational phases 1, 2 and 3 (day 144, 83 and 94, respectively).

**Figure 6 membranes-10-00317-f006:**
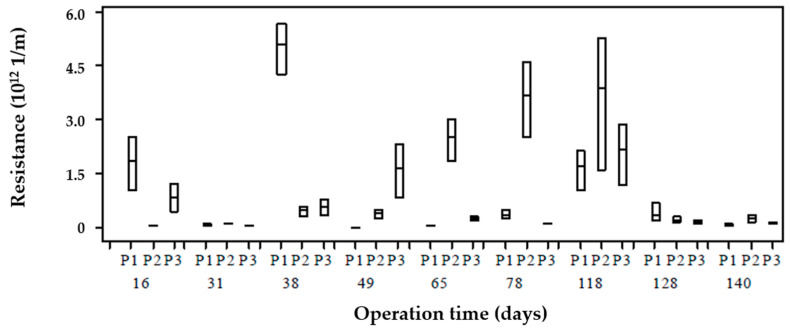
Membrane resistance distribution at each point over operation time.

**Figure 7 membranes-10-00317-f007:**
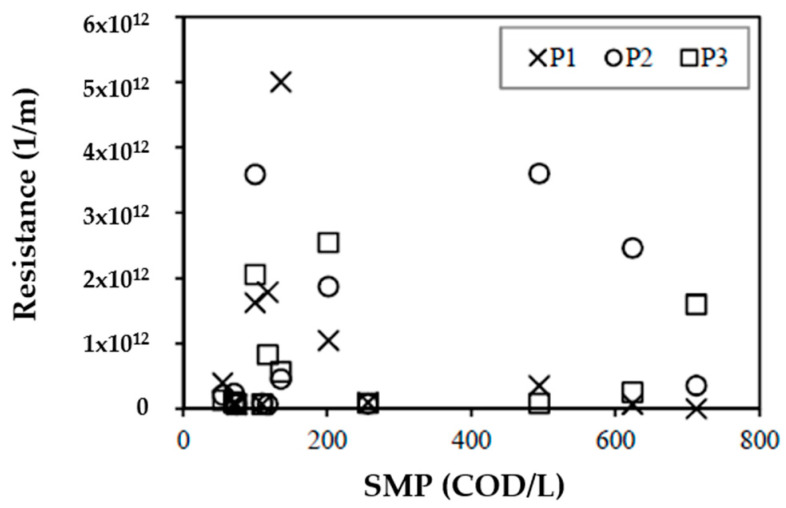
SMP concentration vs. resistance.

**Table 1 membranes-10-00317-t001:** Overall parameter measured in the reactor. Average values ± standard deviation.

	Phase 1	Phase 2	Phase 3
OLR (kg/m^3^d)	3.0 ± 0.8	4.1 ± 0.8	1.2 ± 0.4
pH	7.0 ± 0.2	6.9 ± 0.1	6.9 ± 0.1
**% COD Removal**			
P1	95.9 ± 2.7	94.0 ± 4.4	95.8 ± 3.4
P2	96.9 ± 1.9	93.4 ± 4.9	95.4 ± 4.0
P3	96.5 ± 2.7	93.5 ± 4.3	95.8 ± 3.8
P4	98.7 ± 0.4	97.1 ± 3.4	98.6 ± 0.3
**VSS (% VSS/TSS) ***			
P1 g/L	15.6 ± 3.7 (86.6 ± 2.9)	13.7 ± 6.3 (89.2 ± 1.2)	7.3 ± 1.5 (85.7 ± 3.9)
P2 g/L	11.4 ± 1.7 (86.5 ± 2.4)	12.1 ± 6.0 (88.5 ± 1.4)	5.9 ± 1.3 (85.1 ± 3.9)
P3 g/L	11.1 ± 2.4 (86.7 ± 2.3)	10.8 ± 3.8 (88.8 ± 1.0)	5.3 ± 1.4 (84.7 ± 3.5)

* Values in parentheses are the percentages of VSSs in TSSs.

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
