# Peer review of "SMP Production in an Anaerobic Submerged Membrane Bioreactor (AnMBR) at Different Organic Loading Rates"

_membranes, 2020, doi:10.3390/membranes10110317_

Round 1
Reviewer 1 Report
The manuscript is discussing the impact of organic loading rate on SMP production of ANMBR. The whole study is of interest (even though limited) but missing data and control and there are quite some issue regarding the interpretation and conclusions of the study. Before being consider for publication the authors should better introduce the specific state of the art (SMP production in AnMBR), provide a more in depth analysis of their results and be more critical on the interpretation they made of the results.
Here are my comments:
Abstract:
L30-32: I do not see these results/figures in the document. In general the authors should specify should clearly analyse with dedicated figures what is the removal due to biological degradation in the reactors and rejection by the membrane.
L32-34: Not in agreement with data presented in figure 5. No clear trend regarding composition in MW for all conditions between 1st sample and last sample.
Introduction:
L45: smaller footprint?
L52: they ‘’are’’ classified. Example of some languages issues through the document. Please get the manuscript revised, ideally by a fluent english reviewer.
L56-65: please rewrite to be be clearer and more concise. You jump from one thing to the other and it affects the clarity of the manuscript
L67-72: abbreviations used in equations are too long, please shorten. Also the description of the abbreviations should be better explained.
The whole introduction is talking mostly abot MBR, not AnMBR. This should be rewritten since there is already quite a lot of work made on AnMBR. (aerobic) MBR is not the objective here in this study and maybe the behavior is different regarding SMP and MW repartition.
Which applications are targeted? What kind of problematic you want to solve?
State of the art of impact of OLD is not reviewed in the introduction.
Materials and Methods:
2.2: ultrafiltration is in the supernatant or in the sludge?
2.3: synthetic WW. What kind? Urban, industrial? What kind of industry? OLD are very high for urban WW…what is the common OLD in the targeted application?
L129: hydraulic cleaning? What do you mean?
Eq.4: way too long, looks more like text than equation….
Results:
Table2: higher rejection in P4 than in P1,2,3: please explain and discuss. Is the membrane supposed to reject some of these compounds? Average values? Increase of overall rejection with time and membrane fouling?
‘’Unknown COD’’: you use this term through the document. Not very scientific and convincing. Please refer to the literature to find a more appropriate term and make hypothesis on the composition of this fraction.
L295: a more ‘’proportional’’. Again not very scientific and clear what you mean here
The whole section 3.4 should be rewritten and with more specific analyses: Impact of membrane MW cut off on the anlysis should be introduce. Not only repartition is important bt also the overall content of those fractions and COD in permeate etc should be used for the analysis. I do not see also clear trends that are supporting your say regarding initial and final status in the reactor. The 3 loads seem to behave differently. Formation of a fouling layer on the membrane can be a point but no data to support that, please provide TMP data. Also it would be great to provide data regarding the daily follow-up of reactors during the duration of the tests. Is there an initial stabilization period? Are some of the fraction (highest MW) supposed to be rejected by the membrane? Comparing p4 with P1,P2 and P3 should give you informations. From the data presented, MW distribution is more connected to operation of the reactor, not to membrane rejection or building up of an additional layer, P4 composition seems similar to P1,P2 and P3.
L316: there is no fouling data…
L319-321. I do not know how you conclude that based on the results presented?
Author Response
- L30-32: I do not see these results/figures in the document. In general the authors should specify should clearly analyse with dedicated figures what is the removal due to biological degradation in the reactors and rejection by the membrane.
These values are described in section 3.2, which refers to Figure 3a
- L32-34: Not in agreement with data presented in figure 5. No clear trend regarding composition in MW for all conditions between 1st sample and last sample.
The abstract was modified and the analysis of section 3.4 was modified.
- L45: smaller footprint?
It was corrected.
- L52: they ‘’are’’ classified. Example of some languages issues through the document. Please get the manuscript revised, ideally by a fluent english reviewer.
A native speaker revised it.
- L56-65: please rewrite to be clearer and more concise. You jump from one thing to the other and it affects the clarity of the manuscript
It was modified, as suggested by the reviewer.
- L67-72: abbreviations used in equations are too long, please shorten. Also the description of the abbreviations should be better explained.
Equations and abbreviations were simplified, as suggested by the reviewer.
- The whole introduction is talking mostly about MBR, not AnMBR. This should be rewritten since there is already quite a lot of work made on AnMBR. (aerobic) MBR is not the objective here in this study and maybe the behavior is different regarding SMP and MW repartition.
The introduction was modified, as suggested by the reviewer. Also, it was included in the literature review of the MW distribution of SMP.
- Which applications are targeted? What kind of problematic you want to solve?
The introduction was modified (see comment 7), and the objective was modified in the last paragraph of the introduction.
- State of the art of impact of OLD is not reviewed in the introduction.
It is resolved in the first and fourth paragraphs of the introduction
- 2: ultrafiltration is in the supernatant or in the sludge?
The submerged membrane was located at the top of the reactor. This was added to the methodology section, and the diagram was modified.
- 3: synthetic WW. What kind? Urban, industrial? What kind of industry? OLD are very high for urban WW…what is the common OLD in the targeted application?
In section 2.3, the first paragraph shows the range of OLD to which ANMBRs can be applied, so the study focuses on evaluating SMP in part of this range of the industrial sector OLR (3-9), simulating the carbon source with glucose to gain a greater understanding of the behavior of SMP.
- L129: hydraulic cleaning? What do you mean?
It was corrected; it is the backwash cleaning.
- 4: way too long, looks more like text than equation….
It was modified, as suggested by the reviewer.
- Table2: higher rejection in P4 than in P1,2,3: please explain and discuss. Is the membrane supposed to reject some of these compounds? Average values? Increase of overall rejection with time and membrane fouling?
Section 3.1 was modified to include the reviewer suggestions and other references was included
- ‘’Unknown COD’’: you use this term through the document. Not very scientific and convincing. Please refer to the literature to find a more appropriate term and make hypothesis on the composition of this fraction.
The term Unknown COD is used to represent compounds other than carbohydrates, proteins, and VFA in COD units. This term has been widely used in SMP and EPS and is calculated as described in the methodology section. We have taken this term from the following references https://doi.org/10.1002/jctb.1622 ; https://doi.org/10.1021/acs.est.6b05791 ; doi:10.1016/j.procbio.2014.09.013.
- L295: a more ‘’proportional’’. Again not very scientific and clear what you mean here
The text was adjusted as suggested by the reviewer.
- The whole section 3.4 should be rewritten and with more specific analyses: Impact of membrane MW cut off on the anlysis should be introduce. Not only repartition is important bt also the overall content of those fractions and COD in permeate etc should be used for the analysis. I do not see also clear trends that are supporting your say regarding initial and final status in the reactor. The 3 loads seem to behave differently. Formation of a fouling layer on the membrane can be a point but no data to support that, please provide TMP data. Also it would be great to provide data regarding the daily follow-up of reactors during the duration of the tests. Is there an initial stabilization period? Are some of the fraction (highest MW) supposed to be rejected by the membrane? Comparing p4 with P1,P2 and P3 should give you informations. From the data presented, MW distribution is more connected to operation of the reactor, not to membrane rejection or building up of an additional layer, P4 composition seems similar to P1,P2 and P3.
The section 3.5 was included to further discuss membrane fouling, including data of the membrane resistance over time.
SMP data were analyzed with Minitab 16 software, and it was determined that there were no statistically significant differences (p > 0.05) in the SMP extracted from sludge at different heights (P1, P2, and P3) among the three phases. The analysis of the results obtained was included in the manuscript.
- L316: there is no fouling data…
A new section (3.5) was added to include the fouling data.
- L319-321. I do not know how you conclude that based on the results presented?
That was modified accordingly in the conclusion section.

Reviewer 2 Report
The manuscript membrane-910595 deal with the molecular weight distribution in an upflow AnMBR. The topic is interesting, and comes with related experimental data. However, the data was not completely reported, making the manuscript looks like a piece of fragile. The significant and problem was not clearly define and critically discussed. The manuscript must be significantly improved before it can be published.
Introduction
L41-50 Focus on AnMBR, particularly fouling and SMP role.
L51-65 Please explain the State of Art for SMP in AnMBR fouling.
L66-72 Should be Method.
L72-97 the question should be clear putted.
M&M
31.4L is hardly a pilot. Reactor configuration was not clearly putted.
I am not sure the membrane module illustrate is correct.
Results
The discussion need greatly enhanced with a critical way, comparing with existed works.
The SMP's relationship with fouling and foulant should be a bigger concern, and COD contribution later.
3D-EEM of the foulant will be much better.
Author Response
- The manuscript membrane-910595 deal with the molecular weight distribution in an upflow AnMBR. The topic is interesting, and comes with related experimental data. However, the data was not completely reported, making the manuscript looks like a piece of fragile. The significant and problem was not clearly define and critically discussed. The manuscript must be significantly improved before it can be published.
Base on the comments from Reviewer 1 and 2, we consider that the manuscript was improved considerably.
- L41-50 Focus on AnMBR, particularly fouling and SMP role.
Refer to the answer to comment 7.
- L51-65 Please explain the State of Art for SMP in AnMBR fouling.
Refer to the answer to comment 7.
- L66-72 Should be Method.
It was moved to the materials and methods sections
- L72-97 the question should be clear putted.
Refer to the answer to comment 8.
- 4L is hardly a pilot. Reactor configuration was not clearly putted.
We agreed with the reviewer; it is a lab-scale AnMBR.
- I am not sure the membrane module illustrate is correct.
Figure 1 was modified.
- The discussion need greatly enhanced with a critical way, comparing with existed works.
The results and discussion section were improved based on the reviewers' suggestions.
- The SMP's relationship with fouling and foulant should be a bigger concern, and COD contribution later.
Refer to the answer to 18.
- 3D-EEM of the foulant will be much better.
We agreed with the reviewer; however, in the lab, we do not have that capability for such a test.

Round 2
Reviewer 1 Report
The document has been generally improved but the manuscript remain of limited interest; despite a lot of analysis, the data provided are not fully clear as the trends are not straightforward regarding fouling or SMP size repartition and evolution over time.
The introduction has been highly improved. Still, the authors should specify at the end of the intro what will be the additional and novel contribution of their work and what will be done in their study.
English is still so-so…especially in the corrected sections…
- ‘’Unknown COD’’: you use this term through the document. Not very scientific and convincing. Please refer to the literature to find a more appropriate term and make hypothesis on the composition of this fraction.
The term Unknown COD is used to represent compounds other than carbohydrates, proteins, and VFA in COD units. This term has been widely used in SMP and EPS and is calculated as described in the methodology section. We have taken this term from the following references https://doi.org/10.1002/jctb.1622 ; https://doi.org/10.1021/acs.est.6b05791 ; doi:10.1016/j.procbio.2014.09.013.
Please cite those docs then and a bit of background and what they typically are.
Author Response
- The document has been generally improved, but the manuscript remains of limited interest; despite much analysis, the data provided are not entirely clear as the trends are not straightforward regarding fouling or SMP size repartition and evolution over time.
The results and discussion section was improved, as suggested by the reviewer in lines:
Section 3.1: 199-207, 248-258, 265-269
section 3.3: 278-306,
Section 3.4: 319-353
Section 3.5: 365-408
- The introduction has been highly improved. Still, the authors should specify at the end of the intro what will be the additional and novel contribution of their work and what will be done in their study.
The introduction was modified accordingly, as suggested by the reviewer in line 42-96, and at the end of the section, the research objective was included.
- English is still so-so…especially in the corrected sections…
The English have been revised again on the highlighted sections.
- "Unknown COD": you use this term through the document. Not very scientific and convincing. Please refer to the literature to find a more appropriate term and make a hypothesis on the composition of this fraction. The term Unknown COD is used to represent compounds other than carbohydrates, proteins, and VFA in COD units. This term has been widely used in SMP and EPS and is calculated as described in the methodology section. We have taken this term from the following references https://doi.org/10.1002/jctb.1622; https://doi.org/10.1021/acs.est.6b05791; doi:10.1016/j.procbio.2014.09.013. Please cite those docs then and a bit of background and what they typically are.
Aquino et al. (2006), Characterization of dissolved compounds in submerged anaerobic membrane bioreactors (SAMBRs) https://doi.org/10.1002/jctb.1622
They study SMP on AnMBR and define the unknown COD as follows: "The remaining COD was categorized as 'unknown'; i.e., compounds that did not appear to be carbohydrate, protein or VFA. The results presented show that the percentage of protein in SAMBR 2 (29.8% COD) was lower than in SAMBR 1 (49.8% COD) and this might indicate that these compounds were preferentially adsorbed by powdered activated carbon (PAC); as a result, the fraction of 'unknown' was slightly higher in the supernatant of SAMBR 2."
Kunacheva et al. (2017), Chemical Characterization of Low Molecular Weight (MW < 580 Da)
Soluble Microbial Products (SMPs) in an Anaerobic Membrane Bioreactor https://doi.org/10.1021/acs.est.6b05791
They study low molecular weight SMP on AnMBR, and they define an unknown quantity, a figure is presented here below:
Luna et al. (2014), EPS and SMP dynamics at different heights of a submerged anaerobic membrane bioreactor (SAMBR) https://doi.org/10.1016/j.procbio.2014.09.013
They focus on the study of SMP on an AnMBR: "the composition of SMP (proteins, carbohydrates, unknown COD) at different reactor heights. Most SMP components were unknown."
These three studies, which all used the term of unknown COD, that is why in the manuscript, the term is used, as it was not possible to characterize all the components. Based on that, three new paragraphs were added to explain it better in the introduction section and line 265-275. It is also essential to highlight that the aim of this study was not to characterize that unknown COD fraction.

Reviewer 2 Report
Please consider decreasing the conclusion to the solid supported and widely interested. Less than 150 words suggested.
Author Response
- Please consider decreasing the conclusion to the solid supported and widely interested. Less than 150 words were suggested.
We appreciate the reviewer's comment, and the conclusion was summarized and reduced to 180 words.
